# Heat shock factor 1 directly regulates transsulfuration pathway to promote prostate cancer proliferation and survival

J. Spencer Hauck [1], David Moon[1], Xue Jiang[1], Mu-En Wang[1], Yue Zhao[2], Lingfan Xu[3], Holly Quang [4], William Butler[1], Ming Chen [1], Everardo Macias[1], Xia Gao [4,5], Yiping He[1] & Jiaoti Huang [1✉]

There are limited therapeutic options for patients with advanced prostate cancer (PCa). We previously found that heat shock factor 1 (HSF1) expression is increased in PCa and is an actionable target. In this manuscript, we identify that HSF1 regulates the conversion of homocysteine to cystathionine in the transsulfuration pathway by altering levels of cystathionine-β-synthase (CBS). We find that HSF1 directly binds the *CBS* gene and upregulates *CBS* mRNA levels. Targeting CBS decreases PCa growth and induces tumor cell death while benign prostate cells are largely unaffected. Combined inhibition of HSF1 and CBS results in more pronounced inhibition of PCa cell proliferation and reduction of transsulfuration pathway metabolites. Combination of *HSF1* and *CBS* knockout decreases tumor size for a small cell PCa xenograft mouse model. Our study thus provides new insights into the molecular mechanism of HSF1 function and an effective therapeutic strategy against advanced PCa.

[1] Department of Pathology and Duke Cancer Institute, Duke University School of Medicine, Room 301M, Duke South DUMC 3712, 40 Duke Medicine Circle, Durham, NC 27710, USA. [2] Department of Pathology, College of Basic Medical Sciences, and the First Hospital of China Medical University, No.77 Puhe Road, Shenyang North New Area, 110122 Shenyang, China. [3] Urology Department, First Affiliated Hospital of Anhui Medical University, 218 Jixi Road, 230001 Hefei, China. [4] Children's Nutrition Research Center, Department of Pediatrics, Baylor College of Medicine, 1100 Bates Ave One Baylor Plaza, Houston, TX 77030, USA. [5] Department of Molecular and Cellular Biology, 1100 Bates Ave Baylor College of Medicine, Houston, TX, USA. ✉email: jiaoti.huang@duke.edu

Prostate cancer (PCa) is the most common non-cutaneous cancer for men and the second leading cause of cancer-related mortality[1,2]. Hormonal therapy targeting the androgen receptor is the treatment of choice for advanced and metastatic PCa, but it causes significant side effects[3]. Hormonal therapy is not curative and eventually, all patients develop recurrent disease known as castration-resistant PCa (CRPC)[4]. Additionally, hormonal therapy can induce neuroendocrine differentiation in about seventeen percent of cases leading to the development of small cell neuroendocrine PCa (SCNC), which is rapidly lethal[5,6]. The development of androgen receptor-independent strategies to effectively target CRPC is therefore urgently needed.

Heat Shock factor 1 (HSF1) is a stress response transcription factor that has been identified as a proto-oncogene in many solid tumors including breast, colon, lung, skin, liver, pancreas, myeloma, prostate, and cervix[7–10]. HSF1 is activated in cancer from reactive oxygen species, protein damage due to oxidative stress, mTOR hyper-activation, increased RAS/MAPK signaling, proteomic imbalance from aneuploidy, and destabilized protein folding from various genetic modifications[8,11]. HSF1 has been identified as a biomarker for many solid tumors[12]. HSF1 staining by immunohistochemistry predicted survival in primary PCa patients treated by radical prostatectomy[13]. We previously identified a HSF1 small molecule inhibitor SISU-102 that inhibits the growth of both hormone sensitive and hormone-resistant PCa lines and CRPC mouse models, and this compound is now under commercial development for clinical trials[14]. In the current study, we used this compound in combination with other approaches to probe the molecular mechanisms of HSF1 function and to identify additional vulnerabilities of advanced PCa for the purpose of developing novel therapeutic strategies.

HSF1 regulates many energy metabolism genes[15] to promote glucose and lactate uptake in normal and cancer cells and increased glucose utilization[16,17]. HSF1 promotes an increased rate of glucose uptake and preferential production of lactate, known as Warburg effect, in breast and liver cancer cells by upregulating the lactate dehydrogenase enzyme via direct binding to the lactate dehydrogenase gene promoter[18,19]. In a mouse model of c-MYC driven liver cancer, expression of a dominant negative HSF1 downregulated c-MYC levels and lipogenesis, mitochondrial biogenesis, and glycolysis including hexokinase and lactate dehydrogenase[20]. However, metabolic function of HSF1 in PCa remains poorly understood.

Cancers preferentially consume many metabolites including glucose, nucleotides, glutamine, and methionine[21–24]. Studying the preferred metabolic pathways utilized by cancer cells can identify novel vulnerabilities that can be therapeutically targeted, as shown in our previous studies[25,26]. We discovered that HSF1 upregulates a metabolic enzyme cystathionine-β-synthase (CBS) by directly binding to the CBS gene. We showed that inhibition of CBS is more effective than inhibition of HSF1 in inducing PCa cell death. We report that combination treatment targeting HSF1 and CBS is an effective therapeutic strategy for advanced PCa.

## Results

**HSF1 levels are increased in PCa and predict overall survival of mCRPC patients**. To investigate the role of HSF1 in PCa, we first mined The Cancer Genome Atlas (TCGA) to analyze the expression of HSF1 in benign prostate and primary PCa[27]. Compared to benign controls, primary prostate cancer had higher levels of *HSF1* mRNA (TCGA-PRAD) (Fig. 1a). *HSF1* mRNA further increases from primary PCa to mCRPC (GSE35988, GPL6480 series matrix)[28] (Fig. 1b). Relative to mCRPC, *HSF1* mRNA was not increased in SCNC patients (phs001648.v1.p1)[29]

(Fig. 1c). We found HSF1 protein was also elevated in PCas from genetically engineered mouse models. Compared to wildtype littermates, HSF1 protein levels were higher in the prostate tumor and lymph node metastasis of TRAMP B6;FVB F1 mice[30,31], and B6 Pten; Rb double knockout mouse tumors (Supplementary Fig. 1)[32]. Similar observations were also made in PCa cell lines. Compared to the benign prostate cell line RWPE1, most PCa cell lines assessed had higher levels of HSF1 protein levels particularly in the highly aggressive androgen receptor-negative PC3 cells and the NCI-H660 SCNC line (Fig. 1d). Patients whose tumors had high *HSF1* mRNA had reduced overall survival from the onset of mCRPC (Fig. 1e)[33]. Relative to benign prostatic cores in a tissue microarray, nuclear HSF1 staining, or active HSF1, was significantly increased in primary PCa, CRPC, and SCNC cores (Fig. 1f and g). These results demonstrate that HSF1 is expressed throughout the course of PCa, is higher in more aggressive tumors, and predicts overall survival for patients with aggressive PCa.

**Inhibition of HSF1 decreases transsulfuration pathway metabolites in PCa cells**. To study the role of HSF1 in PCa metabolism, we performed metabolite profiling on hormonal therapy resistant C4-2 cells treated with 2.5 μM SISU-102, a specific HSF-1 inhibitor[14], for 48 h. The metabolite levels were analyzed and segregated by pathways (Fig. 2a). Metabolite enrichment analysis using MetaboAnalyst showed that homocysteine degradation metabolites were significantly enriched by the HSF1 inhibitor treatment (Fig. 2b). The upstream transsulfuration pathway (TSS) metabolite o-phospho-serine was increased, and the TSS metabolites homocysteine and cystathionine were decreased by the HSF1 inhibitor treatment (Fig. 2c). The TSS metabolites were decreased by the HSF1 inhibitor treatment in C4-2 cells (Fig. 2d). PC3 cells were treated with 2.5 and 5 μM SISU-102 for 48 h. There was no decrease in TSS metabolites from SISU-102 treatment in PC3 cells, but metabolite enrichment analysis showed that downstream pathways of the TSS taurine and glutathione were affected (Supplementary Fig. 4a, b). Taurine and glutathione pathway metabolites were decreased from SISU-102 treatment in PC3 cells (Supplementary Fig. 4d, f). Taurine was also decreased by SISU-102 treatment in C4-2 cells, but glutathione levels were unaffected by SISU-102 treatment (Supplementary Fig. 4c, e). The methionine cycle, immediately upstream of the TSS, was unaffected by HSF1 inhibition in C4-2 and PC3 cells (Supplementary Fig. 2a, b and Supplementary Fig. 4a). These data indicate that inhibition of HSF1 specifically altered the TSS not the methionine cycle.

**CBS levels are increased in PCa and inhibition of CBS decreases PCa growth**. CBS is the rate-limiting enzyme of the TSS. We analyzed the expression of CBS in benign and primary PCa tissues. Compared to benign controls, *CBS* mRNA was higher in primary PCa and mCRPC[27,28] (Fig. 3a, b). Relative to mCRPC, *CBS* mRNA was not increased in SCNC[29] (Fig. 3c). In PCa TMAs, CBS cytoplasmic staining was observed in primary PCa, CRPC, and SCNC (Fig. 3d). In PCa cell lines, CBS protein was highest in the LNCaP, C4-2, 22Rv1, and CWRR1 adenocarcinoma lines (Fig. 3e and Supplementary Fig. 6a, b). CBS protein was present in the prostate tumor of 15-week-old TRAMP B6; FVB F1 mice but higher in 19-week-old TRAMP B6; FVB F1 mice, which models aggressive PCa (Supplementary Fig. 1). CBS protein was also detected in 12-week-old B6 Pten knockout mice, which models early-stage, hormone-sensitive PCa (Supplementary Fig. 1).

To study the role of CBS in PC growth, we examined the impact of CBS on prostate cell proliferation. Doxycycline

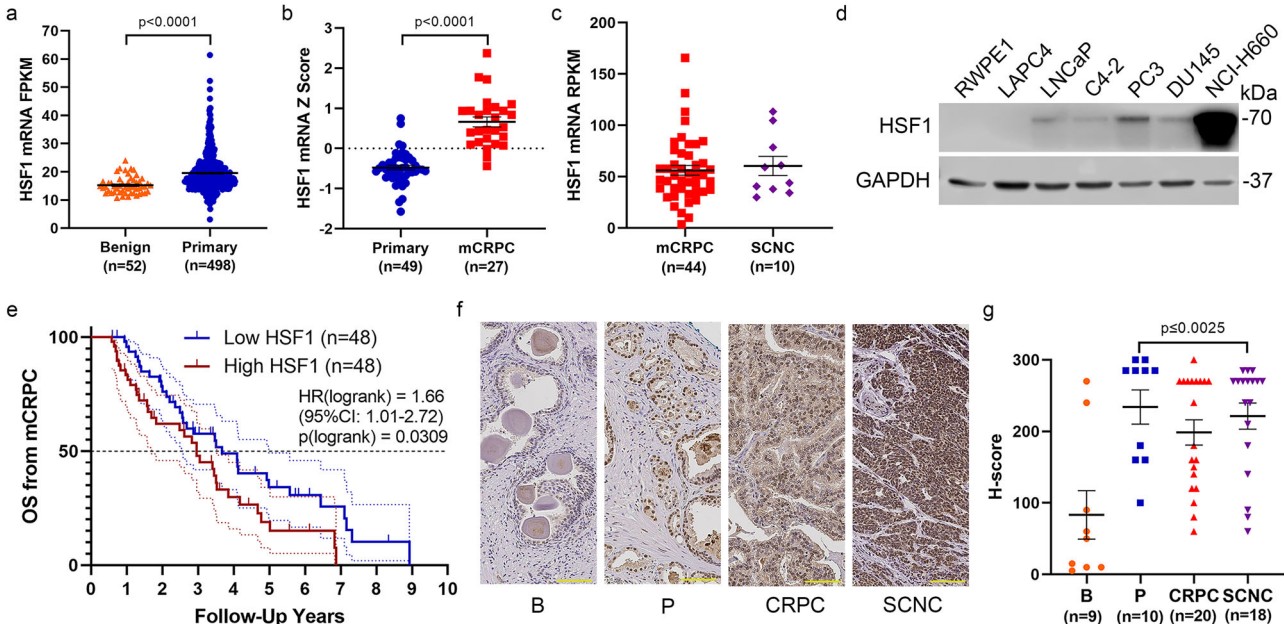

**Fig. 1 HSF1 levels are increased in PCa and predict overall survival of mCRPC patients. a, b** Comparison of *HSF1* mRNA expression between primary tumor (blue, *n* = 498) and benign prostate (orange, *n* = 52) in TCGA database TCGA-PRAD (**a**), mCRPC (red, *n* = 27) and primary tumor (blue, *n* = 49) from GSE35988, GPL6480 series matrix (**b**). **c** Comparison of *HSF1* mRNA in mCRPC (red, *n* = 44) and SCNC (purple, *n* = 10) in phs001648.v1.p1 published data. **d** HSF1 protein levels in different PCa lines. Normal prostate epithelial cell RWPE-1 was used as a control. **e** Overall survival analysis from mCRPC diagnosis of high *HSF1* group (red, *n* = 48) and low *HSF1* group (blue, *n* = 48) in phs001648.v1.p1 published data. Log-rank test, *p* = 0.0309. **f** Representative pictures from immunohistological analysis of HSF1 levels in benign (orange, *n* = 9), primary tumor (blue, *n* = 10), CRPC (red, *n* = 20), and SCNC (purple, *n* = 18). Scale bar= 100 μm. **g** Nuclear localized HSF1 IHC staining was quantified with H-score. Comparison between benign and cancer types is shown. Mean ± standard error is displayed in dot plots. FRKM: fragments per kilobase of transcript per million. RPKM: reads per kilobase transcript. B: benign, P: primary tumor, CRPC: castration-resistant PCa, SCNC: small cell neuroendocrine carcinoma.

inducible knockdown of *CBS* in C4-2 cells decreased growth as well as *CBS* mRNA and protein levels (Fig. 3f, g, i, k). The CBS inhibitor CH004 decreased the cell survival of the C4-2 and PC3 PCa cell lines, but the benign prostate cell lines RWPE1 and BPH-1 were largely unaffected (Fig. 3h, j)[34]. These data indicate that CBS is increased in prostate adenocarcinoma and is an actionable target for PCa.

**Combined targeting of HSF1 and CBS inhibits PCa growth.** Since HSF1 and CBS appeared to be important molecules in a linear pathway, we next assessed the effects of targeting HSF1 and CBS in combination in PCa cells. PCa cells were treated with SISU-102 for 48 h followed by treatment with CBS inhibitor CH004. In C4-2 and PC-3 cells, there was a clear additive effect from SISU-102 and CH004 in a dose-dependent manner (Fig. 4a, b). Moreover, there was an additive effect from inducible *CBS* knockdown in C4-2 cells treated with the HSF1 inhibitor (Supplementary Fig. 6d). Metabolite analysis showed a decrease in the level of homocysteine after the cells were treated with SISU-102, CH004, and their combination (Fig. 4c). There was a greater decrease in the level of cystathionine from treatment with the HSF1 inhibitor, and cystathionine was also decreased from treatment with both the HSF1 and CBS inhibitors (Fig. 4c). A decrease in cystathionine levels was also seen from *HSF1* knockdown alone in C4-2 cells, but the cystathionine levels were more decreased by *CBS* knockdown (Supplementary Fig. 3a, b). Pyruvate levels were decreased most by single CBS inhibitor treatment, and combination of SISU-102 and CH004 decreased pyruvate levels more than SISU-102 treatment alone (Fig. 4c). Compared to either single knockdown or inhibitor treatment, the combination of HSF1 and CBS inhibition or knockdown increased the levels of diphosphoglycerate, 3-phosphoglycerate,

and phosphoenolpyruvate precursors to pyruvate (Supplementary Fig. 3a, d). HSF1 and CBS knockdown or inhibition in C4-2 increased levels of non-essential amino acids including glutamine and glutamate, which are precursors to the tricarboxylic acid (TCA) cycle (Supplementary Fig. 3a, e). These data indicate that the combined targeting of HSF1 and CBS inhibition alter tumor cell metabolism, which contributes to tumor inhibition.

**CBS knockdown or inhibition induces cell death for PCa.** CH004 induced cell death in liver cancer[34]. Thus, we investigated the impact of CH004 on PCa cell lines C4-2 and PC3 cells compared to SISU-102. CH004 induced cell death as detected by Cytotox green binding to the DNA of dead cells in C4-2 and PC3 cells to a much higher degree than SISU-102 (Fig. 5a, b). There was an increase in cell death from *HSF1* knockdown in C4-2, but the level of cell death was much higher from *CBS* knockdown (Fig. 5c). The level of cell death remained high for days after *HSF1* and *CBS* knockdown in C4-2 cells by sequential measurement each day using IncuCyte. Decrease in HSF1 and CBS protein from knockdown in C4-2 cells was verified by western blot analysis (Supplementary Fig. 6c). There was not an additive effect on cell death observed by adding SISU-102 to inducible *CBS* knockdown in C4-2 cells (Supplementary Fig. 6d). These data indicate the additive effect of the HSF1 and CBS inhibitors is due to the effect on metabolites rather than modifying cell death. The combination of HSF1 and CBS inhibitors induced caspase 3 cleavage in the SCNC line NCI-H660 (Fig. 5d). These data suggest that the CBS inhibitor CH004 alone or in combination with the HSF1 inhibitor SISU-102 induced cell death in a wide range of PCa lines representing different stages and histologic types of the disease.

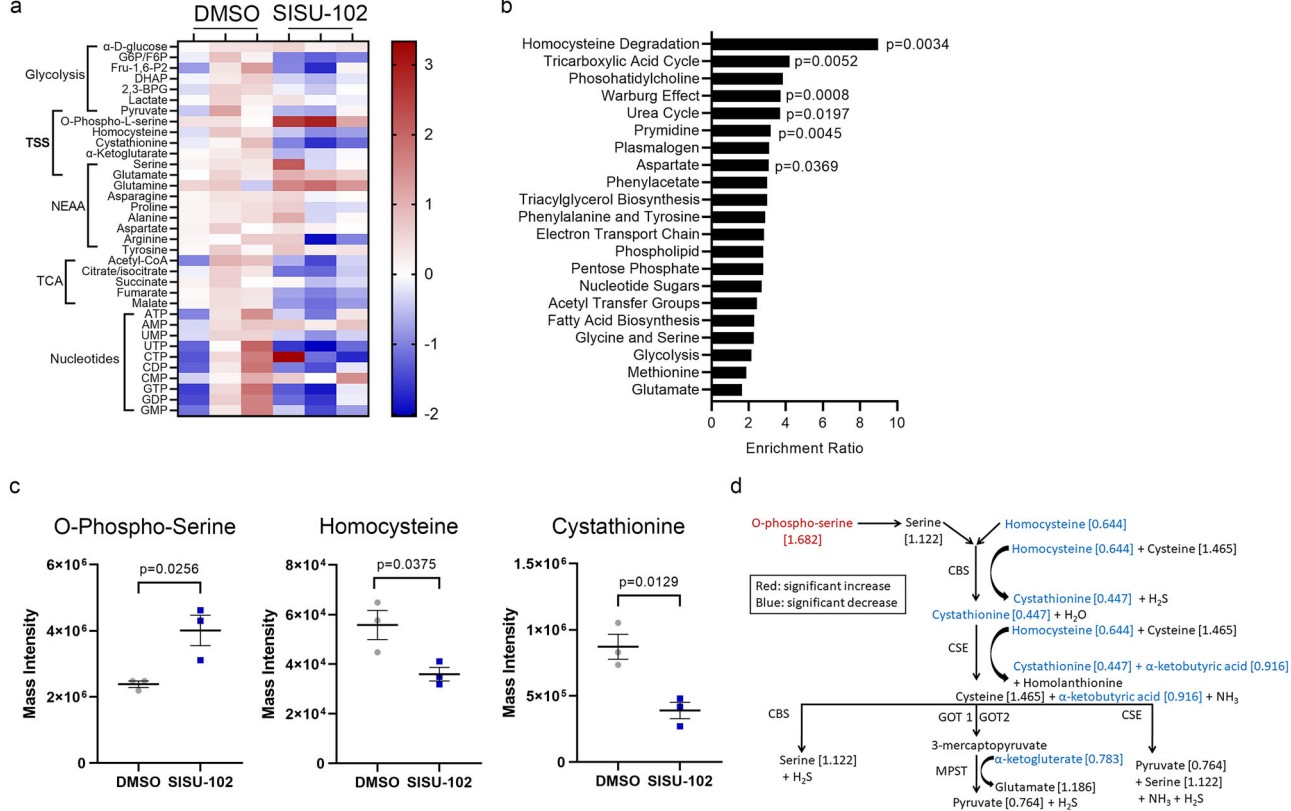

**Fig. 2 Inhibition of HSF1 decreases transsulfuration pathway metabolites in PCa cells. a** Treatment of C4-2 cells with 2.5 SISU-102 for 48 h altered metabolites as shown in the heat map normalized to the average for the DMSO group ($n = 3$). **b** Homocysteine degradation had high enrichment and significance from enrichment analysis with MetaboAnalyst. **c** SISU-102 treatment of C4-2 cells significantly altered o-phospho-serine, homocysteine, and cystathionine levels. **d** Transsulfuration pathway metabolites are shown with red indicating a significant increase in metabolite level and blue indicating a significant decrease in metabolite level. Mean ± standard error is displayed in dot plots. TSS: Transsulfuration pathway, NEAA: non-essential amino acids, and TCA: Tricarboxylic acid cycle.

**HSF1 binding to CBS gene up-regulates CBS mRNA and suppresses PCa cell death.** Since HSF1 inhibition and knockdown decreased cystathionine levels (Fig. 4c and Supplementary Fig. 3b), we tested if HSF1 directly regulates the *CBS* gene[35]. HSF1 binds GAA nucleotides as a homotrimer with a consensus sequence of TTCxxGAAxxTTC where x represents any nucleotide[36]. We identified two putative HSF1 binding sites in the *CBS* gene (Fig. 6a), one in the first intron and the other in the 3' untranslated region (UTR). Analysis of publicly available ChIP datasets revealed that HSF1 binds the *CBS* gene in breast cancer and erythroleukemia cells as shown in Supplementary Table 1[15,37]. *HSF1* knockdown in C4-2 cells led to a decrease in *CBS* mRNA and protein (Fig. 6b–d). *HSF1* knockdown did not affect the other TSS enzyme cystathionine γ-lyase (CTH) (Supplementary Fig. 2c, d). HSF1 ChIP-qPCR showed that in C4-2 cells treated with 2.5 μM SISU-102 for 48 h, there was a decrease in the binding of HSF1 to heat shock response elements in the prompter of HSP70 and the first intron of HSP90 (Fig. 6e). SISU-102 treatment also decreased the binding of HSF1 to both of the putative HSF1 bindings sites in the *CBS* gene (Fig. 6e). As a control, we also found decreased binding of HSF1 to the *CBS* binding sites in C4-2 cells with *HSF1* knockdown (Fig. 6f). We observed a positive correlation of *HSF1* and *CBS* mRNA in mCRPC patients (Fig. 6g). To further validate HSF1's putative binding sites on the *CBS* gene, we utilized an inducible dCas9-HA tagged CRISPR interference assay (CRISPRi) with sgRNAs targeting HSF1 binding sites[38]. HA-tag ChIP-qPCR validated binding of dCas9 at these sites in the CRISPRi system (Fig. 6h).

More importantly, *CBS* mRNA was decreased by CRISPRi at HSF1 sites in C4-2 and 22Rv1 cells, respectively (Fig. 6i, k). An increase in cell death was seen from steric inhibition of HSF1 from binding the *CBS* gene with dCas9 in the CRISPRi assay done in C4-2 and 22Rv1 cells, respectively (Fig. 6j, l). These data indicate that HSF1 binding to the novel binding sites in the *CBS* gene upregulates *CBS* mRNA, which suppressed cell death for PCa cells. Since cysteine levels have been reported to be decreased in the tumor microenvironment[39–42], we performed acute cysteine deprivation experiments in C4-2 and PC3 cells. There was an increase in *CBS* mRNA and protein from 48 h of cysteine deprivation in C4-2 and PC3 cells without an increase in *HSF1* mRNA (Supplementary Fig. 7a–f). There was a decrease in HSF1 binding to both HSF1 binding sites in the *CBS* gene for C4-2 cells from acute cysteine deprivation (Supplementary Fig. 7g).

Since HSF1 regulates the *CBS* gene and CBS is increased in primary PCa, we mined the TCGA database for Gleason score and Grade Group based on *HSF1* and *CBS* mRNA levels[27]. High mRNA levels of *HSF1* or *CBS* alone predicted high Gleason score and Grade Group, but the combination of high *HSF1* and high *CBS* mRNA levels had a greater median than either mRNA alone (Supplementary Fig. 8a, b). The representative images of PCa in different groups based on *HSF1* and *CBS* mRNA showed how these genes have the ability to predict primary PCa aggressiveness in the TCGA database (Supplementary Fig. 8c). The Grade Group median for the *HSF1* high and *CBS* high was 4, which is considered high risk for primary PCa. However, the median for *HSF1* low, *CBS* low was 2, which is considered intermediate

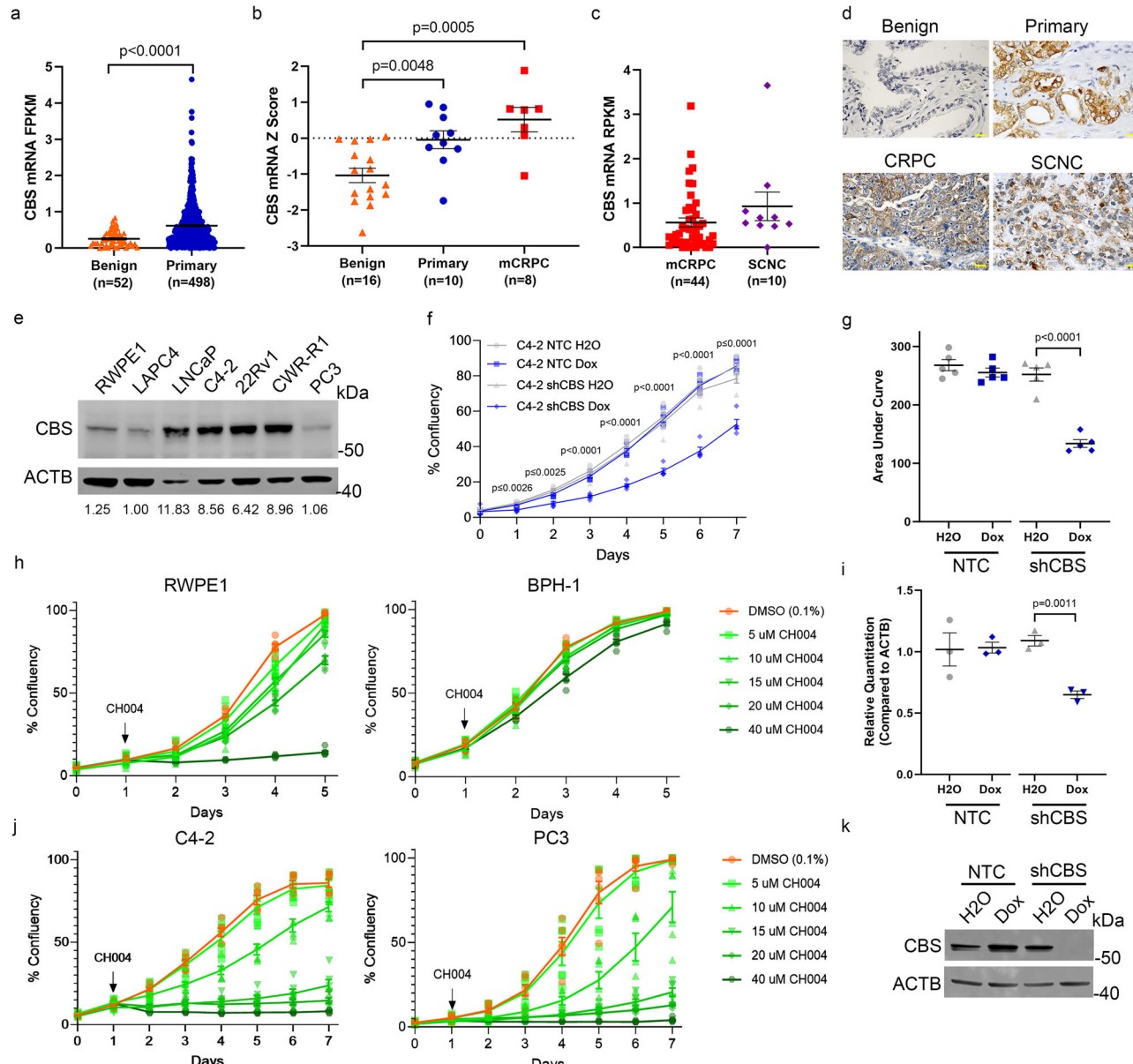

**Fig. 3 CBS levels are increased in PCa and inhibition of CBS decreases PCa growth. a, b** Comparison of *CBS* mRNA expression between primary tumor (blue, *n* = 498) and benign prostate (orange, *n* = 52) in TCGA database TCGA-PRAD (**a**), mCRPC (red, *n* = 8), primary tumor (blue, *n* = 10) and benign prostate (orange, *n* = 16) from GSE35988, GLP6848 series matrix (**b**). **c** Comparison of *CBS* mRNA in mCRPC (red, *n* = 44) and SCNC (purple, *n* = 10) in phs001648.v1.p1 published data. **d** Representative images from immunohistological analysis of CBS levels in benign, primary tumor, CRPC, and SCNC. Scale bar = 25 μm. **e** CBS protein levels in different PCa lines. Normal prostate epithelial cell RWPE-1 was used as a control. The relative level of CBS protein was normalized to LAPC4 with ImageJ. C4-2 with shRNA for *CBS* was treated with 50 ng per mL doxycycline for one week, and growth of the cells were measured with an IncuCyte machine each day (*n* = 5) (**f**). The area under the curve was measured from the first day to the last day of growth (*n* = 5) (**g**). **h, j** The CBS inhibitor CH004 was used to treat the immortalized normal prostate cell line RWPE1 and benign prostate hyperplasia line BPH-1 (*n* = 6) (**h**) and the C4-2 and PC3 PCa lines (**j**) and growth of the cells were measured with an IncuCyte machine each day. The level of *CBS* mRNA (*n* = 3 technical replicates) and protein was evaluated through qPCR and western blot analysis (**i, k**), respectively. Mean ± standard error is displayed in dot plots and line graphs. FRKM: fragments per kilobase of transcript per million. RPKM: reads per kilobase transcript. CRPC: castration-resistant PCa, SCNC: small cell neuroendocrine carcinoma.

favorable for primary PCa[43,44]. These data demonstrate the prognostic power of combined *HSF1* and *CBS* mRNA levels on primary PCa Gleason score and Grade Group.

**Genetic decrease in HSF1 and CBS levels reduce PCa growth.** After the discovery of HSF1 binding sites in the *CBS* gene, we evaluated the effect of targeting *HSF1* and *CBS* on NCI-H660 PCa growth because SCNC is very aggressive and does not respond to

available treatments. Doxycycline-inducible knockout (KO) of both *HSF1* and *CBS* decreased growth for NCI-H660 cells (Fig. 7a) and PC3 cells (Supplementary Fig. 5a). Moreover, exogenous CBS expression in PC3 cells rescued the decrease in growth from treatment with both 2.5 and 5 μM SISU-102 (Supplementary Fig. 5d, e). In a 3D aggregate growth assay, the combination of SISU-102 and CH004 had additive growth inhibitory effects on NCI-H660 cells (Supplementary Fig. 5b). SISU-102 and CH004 also decreased NCI-H660 growth in a cell count

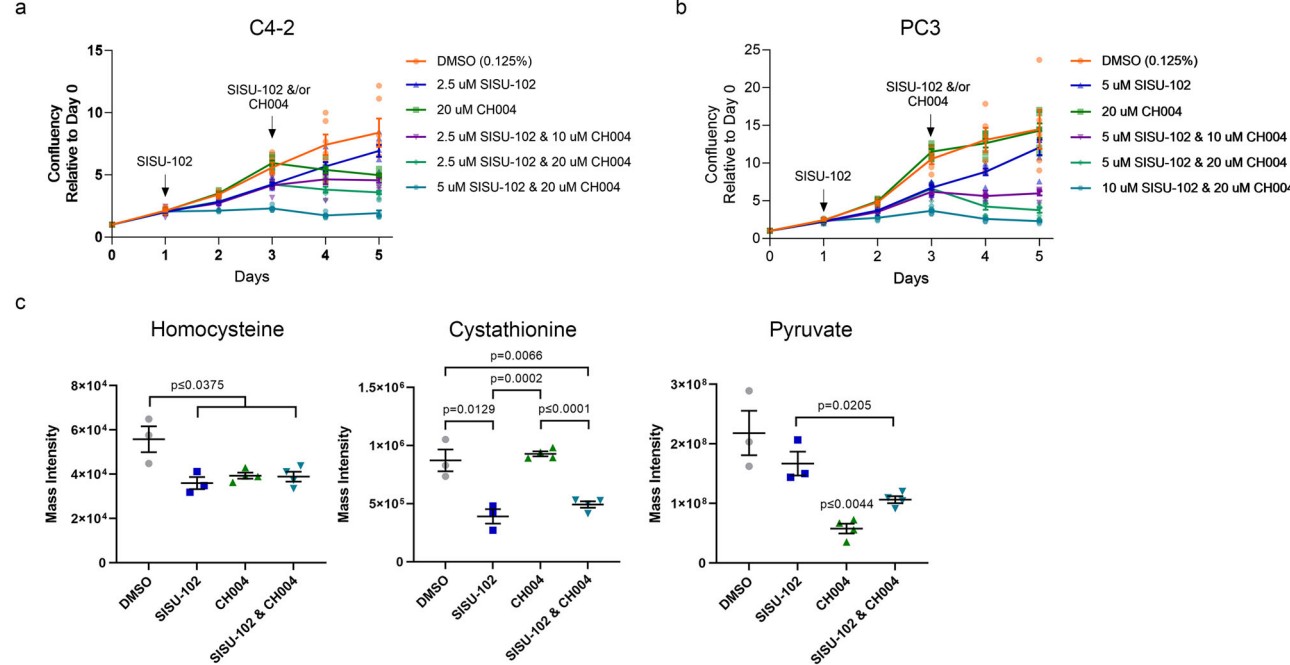

**Fig. 4 Combined targeting of HSF1 and CBS inhibits PCa growth.** To determine if there was an additive effect from inhibiting HSF1 and CBS, the HSF1 inhibitor SISU-102 was used to treat C4-2 (**a**) and PC3 (**b**) cells for 48 h before addition of the CBS inhibitor CH004 because we have shown that the transsulfuration pathway is affected after 48 h of treatment (6 replicates per group). (**c**) C4-2 cells were treated with 2.5 μM SISU-102 or 10 μM CH004 for 48 h before the cells were harvested for measuring homocysteine, cystathionine, and pyruvate metabolite levels ($n = 4$, but 1 replicate was filtered from DMSO and SISU-102 groups due to high variance). Mean ± standard error is displayed in line graphs and dot plots. The p-value CH004 single treatment versus all other groups is shown above CH004 in the pyruvate graph.

assay (Supplementary Fig. 5c). There was not a benefit seen from treating NSG mice bearing NCI-H660 xenografts with 5 mg per kg SISU-102 HSF1 inhibitor (Fig. 7b). However, there was a decrease in tumor size for the *HSF1* and *CBS* double KO NCI-H660 xenograft (Fig. 7c). The knockout of *HSF1* and *CBS* in NCI-H660 xenografts was confirmed by immunohistochemistry (Fig. 7d). We attempted to conduct drug combination studies in vivo, but in our hands CH004 was not well tolerated. There was an initial decrease in NSG mouse weight two days after the initial CH004 treatment (Supplementary Fig. 9a), and 6 of 12 mice treated with CH004 succumbed to adverse effects after treatment with CH004 with or without SISU-102. However, compared to vehicle-treated mice, the CH004-treated mice that survived did not have a decrease in body weight, and there was a decrease in NCI-H660 xenograft size and tumor weight at endpoint in mice treated with two doses of 10 mg per kg CH004 combined with daily 5 mg per kg SISU-102 (Supplementary Fig. 9a). Compared to athymic nude mice treated with daily 5 mg per kg SISU-102, a single dose of 2.5 mg per kg CH004 in nude mice decreased tumor size, and there was a trend of decreased tumor weight at the endpoint (Supplementary Fig. 9b). These data show that *HSF1* and *CBS* knockout and inhibition are able to decrease the growth of the aggressive SCNC cell line NCI-H660 in vivo.

## Discussion

We have shown that benign prostate is negative for HSF1, and HSF1 expression is elevated in primary PCa and remains high throughout the course of PCa including in SCNC. HSF1 is only required in adults for sperm gametogenesis[45], making HSF1 an attractive target for PCa. While the function of HSF1 in supporting PCa cells is well established, its inhibition, although effective, will most likely eventually lose efficacy given tumor cells' adaptive ability. Thus, it is important to dissect HSF's

downstream pathways to identify additional vulnerabilities in PCa cells.

The TSS directly follows the methionine cycle to convert complex amino acids into the high-energy molecule pyruvate[46]. TSS is activated by oxidative stress, which is often increased in PCa[47]. Future studies should evaluate the role of hypoxia, oxidative stress, and reactive oxygen species in the regulation of the *CBS* gene by HSF1. Previous work has shown that activated methionine cycle promotes PCa growth[21,24,48]. Interestingly, s-adenosyl-l-methionine of the methionine cycle promotes CBS activity[49,50], which indicates that inhibition of the methionine cycle also inhibits CBS activity. Loss of *CBS* can lead to homocysteinemia a pathological condition due to the accumulation of homocysteine in the blood which can cause vascular disease[51], but knockout of *CBS* in adult mice results in little pathology[52]. CBS is the rate-limiting enzyme of the TSS[53], and CBS activity produces the antioxidant hydrogen sulfide[24]. The tight regulation of the *CBS* gene by HSF1 likely acts to regulate the level of hydrogen sulfide in addition to metabolites. Our data demonstrates that the effect of HSF1 on the TSS is specific to CBS but not the other TSS enzyme CTH (Supplementary Fig. 2c, d).

CBS has not been widely studied in cancer. However, cysteine deprivation was shown to increase the dependency of cancer cells on the activity of CBS[42], and CBS has been shown to promote ovarian and colon cancer[46,54,55]. Consistent with our data, LNCaP cells were shown to have higher levels of CBS than the being prostate cell line RWPE1[56]. CBS was a viable pharmacological target in liver cancer and chronic myeloid leukemia[34,57]. However, CBS has not been explored as a driver or target in PCa, and the interaction of HSF1 and CBS has not been previously reported. CBS is an exciting therapeutic target because it regulates cancer cell survival. The benign prostate cell lines RWPE1 and BPH-1 were less affected by the CBS inhibitor CH004 than C4-2 and PC3 cells (Fig. 3j, k). RWPE1 was more affected by CH004 than BPH-1

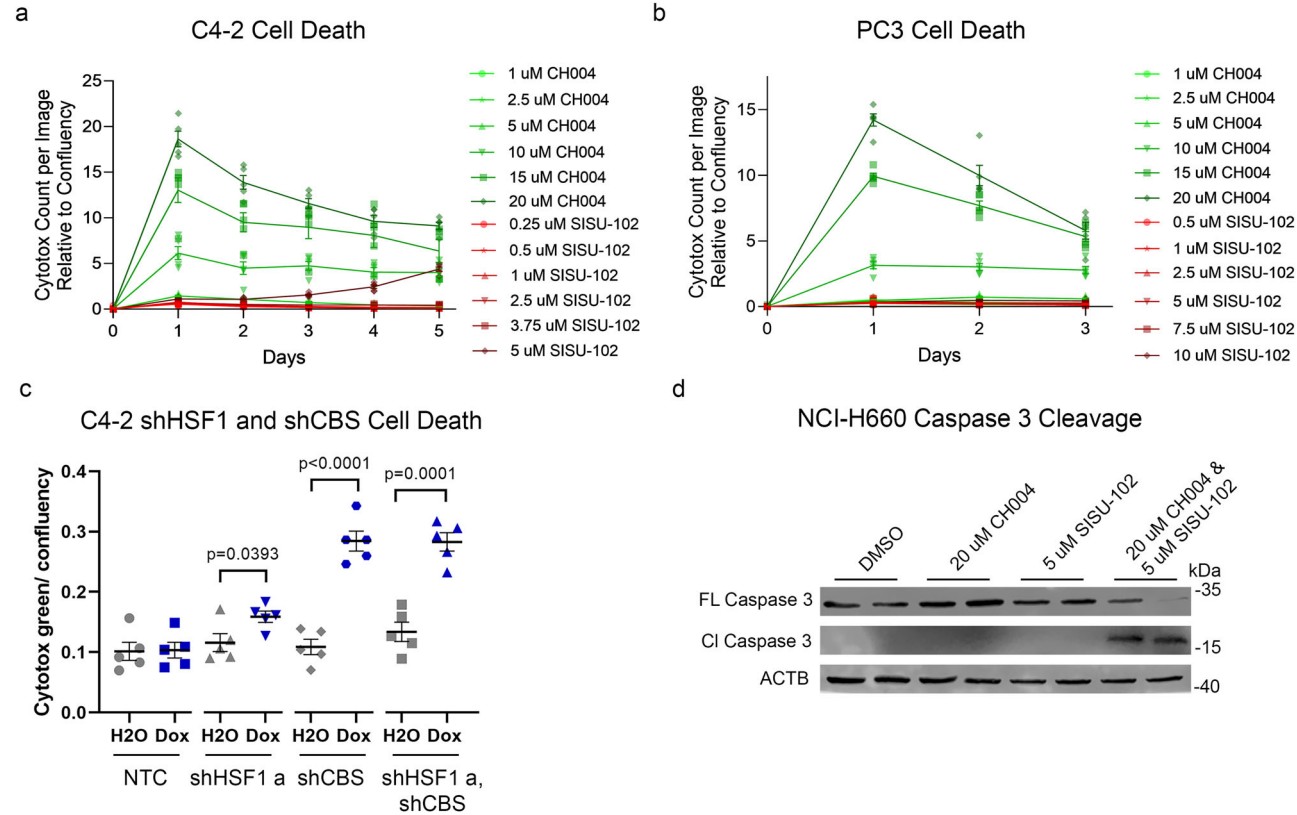

**Fig. 5 CBS knockdown or inhibition induces cell death for PCa.** Cell death was measured in an IncuCyte machine with Cytotox green. The green fluorescence levels were used to analyze the amount of cell death from SISU-102 and CH004 treatment in C4-2 (**a**) and PC3 (**b**) cells ($n = 6$). **c** Cell death was measured each day with Cytotox green in C4-2 cells with knockdown of *HSF1* and *CBS* from treatment with 50 ng per mL doxycycline ($n = 5$). **d** Cell death was measured by caspase 3 cleavage in NCI-H660 cells by western blot analysis ($n = 2$). Mean ± standard error is displayed in dot plots and line graphs.

because it had more CBS protein (Supplementary Fig, 6b). PC3 had low levels of CBS protein, but there was still a decrease in growth after CBS inhibitor treatment and CBS knockout. Low basal levels of CBS may be important for cancer cells since CBS regulates cellular stress response[58]. The benign prostate line RWPE1 had the same or higher level of CBS protein than PC3 (Fig. 3e and Supplementary Fig, 6a, b), but the PC3 cell line was more affected by CH004 than RWPE1, which indicates that CBS was more critical for PCa growth than for benign prostate cells. CH004 and *CBS* knockdown alone induced more cell death than SISU-102 or *HSF1* knockdown in PCa cell lines (Fig. 5a–c). However, only the combination of SISU-102 and CH004 induced caspase 3 cleavage in NCI-H660 (Fig. 5d). There was a clear additive effect on PCa growth from HSF1 and CBS inhibition, but the increase in cell death was primarily from inhibition of CBS.

Unbiased metabolite profiling showed that the TSS was globally decreased from HSF1 inhibitor treatment in C4-2 cells. TSS metabolites levels were not altered from SISU-102 in PC3 cells, but taurine, an amino acid antioxidant, and glutathione, the most abundant antioxidant in mammalian cells, were decreased, indicating there was less flux through the TSS in PC3 cells treated with HSF1 inhibitor (Supplementary Fig. 4a, b, d, f)[59]. C4-2 cells also had a decrease in taurine levels, but there was not a decrease in glutathione levels from SISU-102 treatment (Supplementary Fig. 4c, e). We previously identified that C4-2 cells are more sensitive to SISU-102 than PC3 cells[14]. These data suggest that PC3 cells can compensate for SISU-102 treatment by decreasing the levels of glutathione metabolites in order to maintain steady levels of transsulfuration pathway metabolites. The combination of SISU-102 and CH004 treatment of C4-2 cells decreased the

levels of cystathionine and pyruvate (Fig. 4c). Compared to *HSF1* knockdown in C4-2 cells, *CBS* knockdown decreased cystathionine levels which remained low from the combination of *HSF1* and *CBS* knockdown (Supplementary Fig. 3b). CH004 treatment of C4-2 cells decreased pyruvate levels, but pyruvate levels were not decreased after CBS knockdown in C4-2 cells. Since HSF1 and CBS inhibitor treatment was for 48 h while doxycycline treatment to knockdown *HSF1* and *CBS* was for 7 days, the metabolism data may not be directly comparable. However, *HSF1* and *CBS* knockdown or inhibition increased diphosphoglycerate, 3-phosphoglycerate, and phosphoenolpyruvate precursors to pyruvate levels, and nearly all glycolysis and TCA cycle metabolites analyzed were increased (Supplementary Fig. 3a), indicating that overall carbon utilization was greatly decreased from *CBS* and *HSF1* and *CBS* knockdown in PCa. Reductions in glycolysis and TCA cycle oxidative phosphorylation have been previously reported from CBS inhibition or silencing in colon and ovarian cancer[54,60]. Consistent with previous reports, HSF1 inhibitor treatment of C4-2 decreased the level of glycolysis and TCA metabolites (Fig. 2a)[16,17]. Interestingly, there was an increase in glutamine and glutamate from both HSF1 and CBS knockdown and inhibition (Supplementary Fig. 3e). We have previously shown that PCa is addicted to glutamine to fuel the TCA cycle and pyrimidine metabolism[25,26]. TSS regulates glutamate levels through the action of glutamate oxaloacetate transaminase 1[58], but TCA metabolites and many nonessential amino acids were increased from *CBS* knockdown alone and *HSF1* and *CBS* knockdown in C4-2 cells (Supplementary Fig. 3a), indicating that oxidative phosphorylation and nonessential amino acids were globally affected. The increase in the level of glutamine and

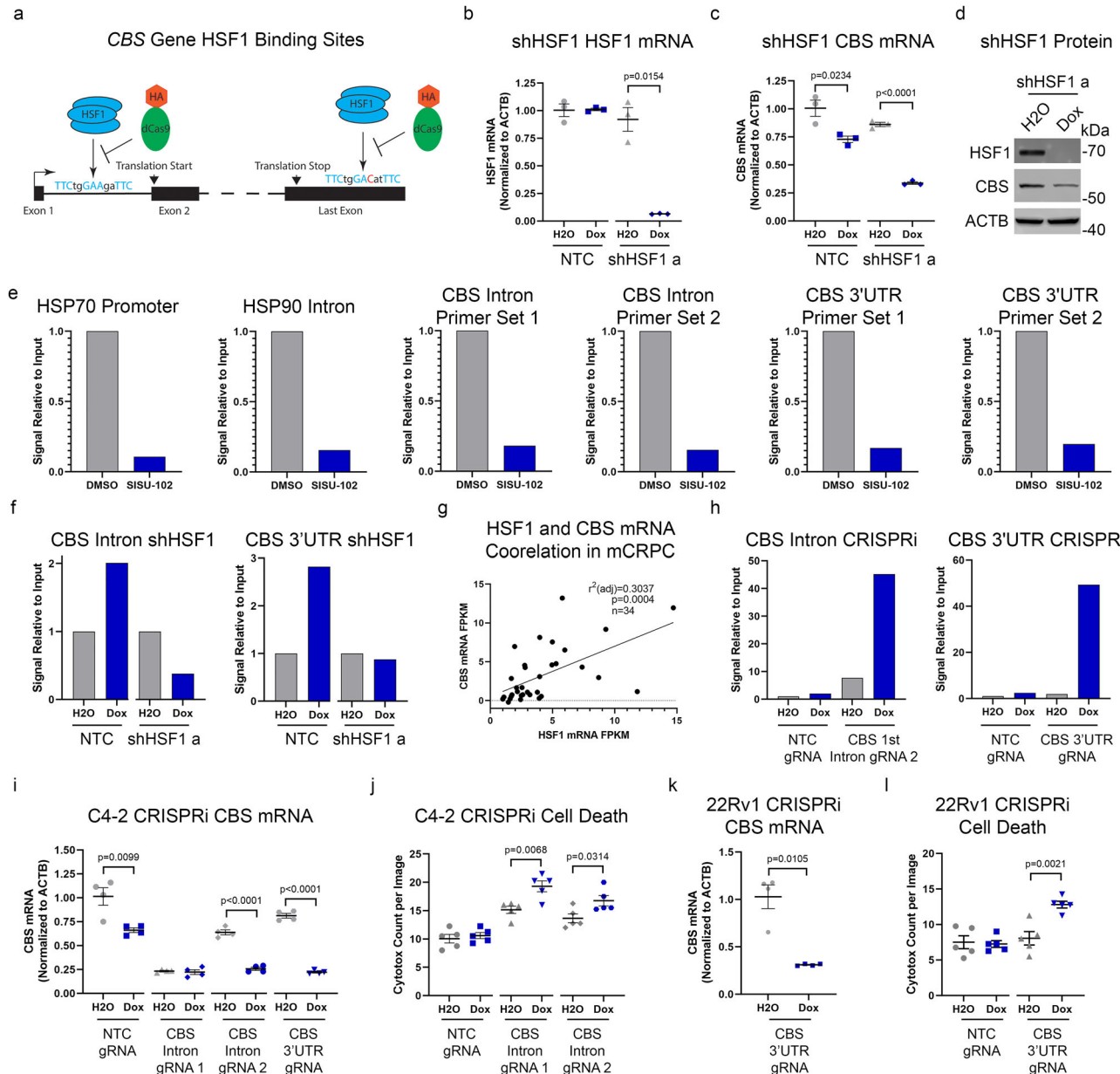

**Fig. 6 HSF1 binding to *CBS* gene up-regulates *CBS* mRNA and suppresses PCa cell death. a** HSF1 binding sites in the Human *CBS* gene are shown with matches to the HSF1 consensus binding site shown in blue and mismatches shown in red. The HSF1 protein binds the DNA unless a deactivated Cas9 (dCas9 with HA tag in orange) targets the HSF1 binding sites in CRISPRi assay. **b**, **c** C4-2 cells with an inducible shRNA targeting *HSF1* were treated for 7 days with 50 ng per mL doxycycline and *HSF1* (**b**) or *CBS* (**c**) mRNA levels were measured ($n = 3$ technical replicates). **d** C4-2 cells with an inducible shRNA targeting HSF1 were treated for 4 days with 50 ng per mL doxycycline and HSF1 and CBS protein levels were evaluated with western blot analysis. **e** C4-2 cells were treated with 2.5 µM SISU-102 for 48 h and HSF1 binding to DNA was analyzed by ChIP-qPCR with HSP70 and HSP90 for controls and two primers for each of the HSF1 binding sites in the *CBS* gene ($n = 4$ technical replicates). **f** C4-2 cells with an inducible shRNA targeting *HSF1* were treated for 7 days with 50 ng per mL doxycycline and HSF1 binding to DNA was analyzed by ChIP-qPCR with primers for each of the HSF1 binding sites in the *CBS* gene ($n = 4$ technical replicates). **g** *HSF1* and *CBS* mRNA has a positive relationship in mCRPC patients from the phs000909.v1.p1 dataset with a p-value of $p = 0.0004$ ($n = 34$). **h** C4-2 cells were transduced with a doxycycline inducible deactivated Cas9 CRISPR interference system (CRISPRi) that targeted each of the HSF1 binding sites in the *CBS* gene. The cells were treated with 250 ng/mL doxycycline for three days, and the biding of the dCas9 was analyzed with by ChIP-qPCR with primers for each of the HSF1 binding sites in the *CBS* gene ($n = 4$ technical replicates). **i-l** *CBS* mRNA levels and cell death were measured after treatment with 250 ng per mL doxycycline for three days with the dCas9 system in C4-2 (**i**, **j**) and 22Rv1 (**k**, **l**) cells. *CBS* mRNA levels were normalized to *ACTB* after CRISPRi ($n = 4$ technical replicates) (**i**, **k**). Cell death was measured with Cytotox green after CRISPRi ($n = 5$) (**j**, **l**). Mean ± standard error is displayed in dot plots. The mean is only displayed in bar graphs for ChIP-qPCR experiments.

glutamate metabolites further supports that the additive effect of inactivating HSF1 and CBS is due to altered cellular metabolism. The linkage of duel HSF1 and CBS inhibition on increasing levels of precursors to pyruvate and glutamine and glutamate should be further investigated.

The HSF1 binding sites in the *CBS* gene are likely required for constitutive expression of *CBS* mRNA since knockdown of *HSF1* decreases *CBS* mRNA and protein, and CRISPRi assay decreased *CBS* mRNA. Moreover, HSP90 is highly regulated by HSF1, and there are HSF1 binding sites in the first intron of the *HSP90* gene,

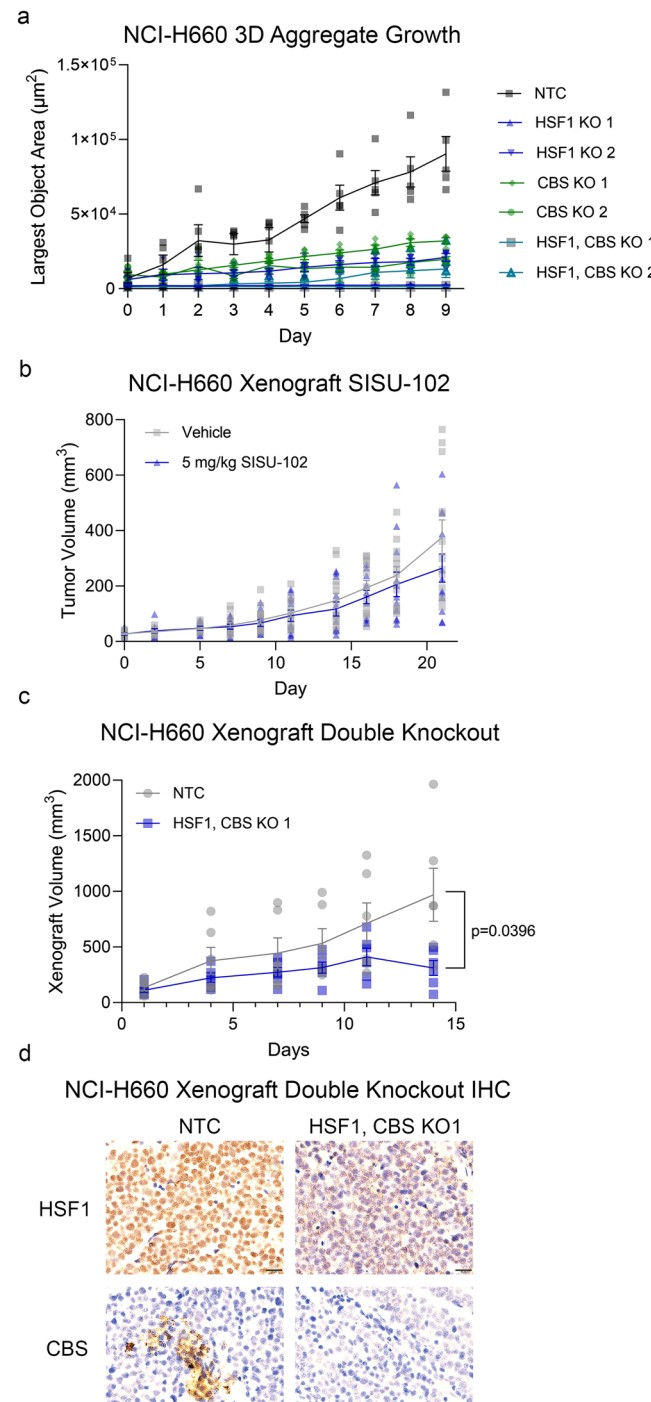

**Fig. 7 Genetic decrease in *HSF1* and *CBS* levels reduce PCa growth.** NCI-H660 cells with a doxycycline-inducible Cas9 and two guide RNAs targeting either the *HSF1* and *CBS* genes were treated with 1 µg per mL doxycycline for twelve days before plating into a 3D aggregate growth assay without doxycycline, and growth of the cells were measured with an IncuCyte machine each day ($n = 5$) (**a**). **b** SISU-102 treatment does not decrease growth for NCI-H660 xenograft. (Vehicle $n = 9$, SISU-102 $n = 10$) **c**, **d** NCI-H660 HSF1 and CBS KO 1 cells were transplanted into NSG mice and treated with doxycycline chow. The xenograft size was measured over the course of the treatment ($n = 6$) (**c**). The tumors were analyzed with immunohistochemistry to evaluate the protein level of HSF1 and CBS (**d**). Scale bar = 20 µm. Mean ± standard error is displayed in dot plots and line graphs.

which are required for constitutive expression of HSP90[61]. The 3'UTR of genes often regulate mRNA stability. Both of the identified HSF1 binding sites positively regulate *CBS* mRNA. The first intronic HSF1 binding site appears to be more tightly bound by HSF1 than the 3'UTR site as shown by the ChIP-qPCR data with the *HSF1* knockdown (Fig. 6f), which is consistent with the mismatch in the 3'UTR site with the consensus HSF1 binding site while the first intron site is a perfect match (Fig. 6a). There appeared to be some leakiness of dCas9 expression in the CRISPRi system because the first gRNA for the CBS intron had decreased *CBS* mRNA levels without doxycycline treatment (Fig. 6i). While no study has directly evaluated the link of HSF1 and CBS, analysis of public datasets revealed that HSF1 regulates *CBS* mRNA levels[15,62]. CBS and HSF1 protein levels did not correlate in our protein analysis of PCa cell lines, possibly because there are many mechanisms by which protein levels can be regulated for these proteins including stress response[8,11,58]. Cysteine deprivation increased CBS mRNA and protein levels in C4-2 and PC3 cells, while there was a decrease in HSF1 binding to the *CBS* gene (Supplementary Fig. 7). These data suggest that CBS can be upregulated by cysteine deprivation independent of HSF1 activity. Future studies should investigate the regulation of *CBS* mRNA by HSF1 in PCa with cysteine levels and culture conditions that mimic the PCa tumor microenvironment. Interestingly, unbiased metabolite profiling of 262 clinical PCa samples showed elevated levels of cysteine, homocysteine, and cystathionine in metastatic PCa and primary PCa compared to benign tissues[63], and increased serum levels of cysteine, homocysteine, and cystathionine independently predicted risk of early biochemical PCa recurrence and aggressiveness[63,64].

SCNC patients had high HSF1 but lower CBS protein expression. However, the SCNC line NCI-H660 had caspase 3 cleavage from the combination of the HSF1 and CBS inhibitors. There was not a benefit seen from treating NSG mice bearing NCI-H660 xenografts with 5 mg per kg SISU-102 HSF1 inhibitor, a dose that greatly decreases the in vivo growth of C4-2, 22Rv1, and TRAMP-C2 tumors[14]. NCI-H660 has a very high level of HSF1 protein, so this level of SISU-102 treatment may not have been sufficient to inhibit the function of HSF1 in NCI-H660 cells. However, *HSF1* and *CBS* knockout decreased NCI-H660 growth in vivo. NCI-H660 was selected to evaluate the effect of targeting HSF1 and CBS because we have previously shown that less aggressive in vivo models of prostate cancer including C4-2, 22Rv1, and TRAMP-C2 have a strong response to HSF1 inhibition alone[14]. Since the vast majority of SCNC biopsies are taken from metastatic sites, future studies should evaluate the efficacy of HSF1 and CBS inhibition in metastatic neuroendocrine PCa[65]. We believe that the benefit of HSF1 and CBS knockdown or inhibition in NCI-H660 was from two hits in the TSS that decrease metabolites better than either treatment or knockout alone, and combined inhibition of HSF1 and CBS represents an effective therapeutic strategy for advanced PCa.

## Methods

**Cell lines, plasmids, and reagents**. Human PCa lines, LNCAP, C4-2, PC3, DU145, 22Rv1, NCI-H660, and immortalized prostatic epithelial cell line RWPE1 were purchased from ATCC. Benign prostatic hyperplasia BPH-1 cells were purchased from Millipore Sigma. Human PCa line LAPC4 was obtained from the laboratory of Charles Sawyers[66] and CWR-R1 was obtained from the laboratory of Elizabeth M. Wilson[67]. BPH-1, LNCaP, C4-2, 22Rv1, CWR-R1, PC3, and DU145 cells were maintained in RPMI-1640 with 300 mg/L L-glutamine and 65 mg per L cysteine dihydrochloride medium (Gibco 11875) and LAPC4 were maintained IMDM (Gibco 12440) supplemented with 10% fetal

bovine serum (Gibco 10437) and 1x streptomycin and penicillin (Genesee 25-512). RWPE1 cells were maintained in keratinocyte serum-free media (Gibco 17005042). NCI-H660 cells were maintained in stem media: Advanced DMEM/F-12 (Thermo-Fisher 12634010) supplemented with 1X B-27 supplement (Gibco 17504044), 10 ng per mL FGF-2 (PeproTech AF-100-18B), 10 ng per mL EGF (PeproTech AF-100-15), 1X Glutamax (Thermo-Fisher 35050061) and 1x streptomycin and penicillin (Genesee 25-512). For cysteine deprivation experiments, RPMI 1640 media without methionine, cystine and L-glutamine (Millipore Sigma R7513) was purchased and supplemented with 15 mg per L methionine (Sigma M5308), 300 mg per L glutamine (Sigma G8540), 10% fetal bovine serum (Gibco 10437) and 1x strepto-mycin and penicillin (Genesee 25-512). The media was further supplemented with L-cysteine dihydrochloride (Sigma L-C6727) to 64, 32,16, 8, or 0 mg per L. C4-2 and PC3 cells were washed with PBS prior to cysteine deprivation and treated with cysteine deprivation media for 48 h before RNA, protein, or ChIP-qPCR analysis. Sigma Mission shRNAs were cloned into the Tet-pLKO-puro (Addgene 21915) or Tet-pLKO-neo (Addgene 21916) vec-tors and compared to the pLKO-Tet-On-shRNA-Control (Addgene 98398) and are shown in Supplementary Table 2. The dCas9 vector p-LV-TRE3G-dCas9-DsRed-Zeo with a HA tag was supplied by S. Ali Shariati at University of California Santa Cruz[38], and guide RNAs targeting the HSF1 binding sites in the CBS gene were cloned into pLV-U6-PGK-Puro by the Duke Viral Vector Core and are in Supplementary Table 2. The HSF1 and CBS knockout gRNA designed by Origene (KN200314 for HSF1, KN401755 for CBS) were cloned into the doxycycline-inducible pCW-Cas9 vector[68] (Addgene 50661) are shown in Supplemen-tary Table 2. Knockdown and knockout PCa cells were selected for transduction with 1 µg per mL puromycin (Thermo A1113803) or 300 ng per mL puromycin for NCI-H660 or 400 µg per mL Geneticin (Gibo 10131027). The CBS open reading frame (NM_000071.3) was overexpressed in the doxycycline-inducible pSTV2.0 GFP vector (GenScript Final Construct Name: CBS OHu26151C_pSTV 2.0), and cloning primers are shown in Supplementary Table 2. PC3 cells were transformed with the CBS overexpression and control lentivirus and selected by FACS sorting for GFP. Doxycycline hyclate (Sigma D9891) was pur-chased from Sigma. The HSF1 inhibitor SISU-102 was developed with collaboration with Dennis Thiele and was provided by Sisu Pharma. The previously published CBS inhibitor CH0004[34] was made by David Gooden at the Duke Small Molecule Core.

**Cell growth and viability.** Cells were seeded into a 96-well plate at 5% confluency, approximately 2,500 cells per well, with at least 5 biological replicates. NCI-H660 cells were seeded into an Ultra-Low Attachment Surface Spheroid Microplate (Corning 4520) for 3D aggregate growth with five biological replicates. The growth of the cells was measured daily by the increase in confluency on an IncuCyte S3 Live-Cell Analysis system (Essen Bioscience 4763) with 4 images per well taken with the 10x objective. For cell counts NCI-H660 cells were seeded with $3 \times 10^5$ cells per 6 well plate wells with 3 biological replicates. Cells were counted with a countess II system (ThermoFisher) with $n = 2$ technical repli-cates. The growth of the cells was analyzed by area under the curve by GraphPad Prism that started from the first day of growth to the completion of the experiment. IncuCyte Cytotox Dye (Sartorius 4633) was added to wells at 1:8000 and the green florescence as an indicator of cell death was measured daily to the end of the experiment. Cell death was either analyzed for a spe-cific day or for the entire experiment by area under the curve analysis with GraphPad Prism. IC50 values were measured by treating C4-2 and PC3 cells with a range of SISU-102 and

evaluating the percent confluency for the cells for a given day. The IC50 calculations were performed by GraphPad Prism.

**Metabolite profiling.** C4-2 or PC3 cells were plated into 6 well plates at 35% confluency with 4 technical replicates and treated with 2.5 or 5 µM SISU-102 and 10 µM CH004 for 48 h. C4-2 HSF1 and CBS inducible knockdown cells were plated into 6 well plates at 7% confluency with 4 replicates and treated with 50 ng per mL doxycycline for 7 days. The confluency of each of the wells was measured by the IncuCyte S3 Live-Cell Analysis system (Essen Bioscience 4763) before metabolite isolation. The media was aspirated and 80% pre-chilled HPLC-grade methanol was added to each well. The cells were scrapped into the methanol to extract the metabolites. Cellular debris was removed by cen-trifugation. The supernatant was dried with a speed vacuum and dry pellets were collected and reconstituted for LC/MS analysis as previously described[25,26]. Metabolite enrichment analysis was performed with MetaboAnalyst.

**Tissue microarrays.** Tissue microarrays were constructed as previously reported[25,69,70]. All ethical regulations relevant to human research participants were followed. Benign and adeno-carcinoma prostatectomy specimens from ($n = 40$) patients were used to make the tissue microarrays including adjacent benign tissue from adenocarcinoma patients. CRPC tissue microarrays samples were obtained through transurethral resection patients who received hormonal therapy, rather than prostatectomy, and had urinary obstruction due to tumor recurrence. SCNC tissue microarrays were constructed from ($n = 17$) primary SCNC cases. All samples were collected from patients with informed consent, and all related procedures were performed with the approval of the internal review and ethics boards of Duke Uni-versity. Androgen receptor was uniformly positive in adeno-carcinoma samples and negative in SCNC samples. Immunohistochemistry was performed following standard pro-cedures with HSF1 (AbCam #ab52757 at 1:200) with EnVision+ Single Reagent (Agilent Dako K400311-2) and CBS (Sigma 3E1 #WH0000875M1 at 200 ng per mL) with EnVision®+ Dual Link System-HRP (DAB+) (Agilent Dako K4065). H-score was quantified by a blinded pathologist.

**Western blot.** Whole-cell lysates were prepared with phosphatase and protease inhibitor cocktail and loaded to 12% resolving SDS-PAGE gels with equal amount of total protein. After electro-phoresis, proteins were transferred to polyvinylidene difluoride transfer membrane (PVDF) followed by blocking in 5% non-fat milk and incubated with primary antibodies. After washing with TBST (TBS with 0.1% Tween), secondary antibodies conjugated with HRP were used to incubate the membranes. Samples were developed by Chemiluminescent Substrate (Thermo Fisher) and exposed by Odyssey Imaging Systems (LI-COR). Primary anti-bodies used are as following: HSF1 (Enzo 10H8 #ADI-SPA-950-D 1: 1000), GAPDH (Cell Signaling 14C10 #2118 1:1000), CBS (Sigma 3E1 #WH0000875M1 1 ug per mL), β-actin (Cell Sig-naling 13E5 #4970 1:1000), Caspase 3 (Cell Signaling #9662 1:1000), and cleaved Caspase 3 (Cell Signaling #9661 1:1000). Secondary antibodies used were Goat anti-rat HRP (Invitrogen # 31470 1:10000), Goat anti-mouse HRP (BioRad #1721011 1.34:4000), and Goat anti-rabbit HRP (BioRad # 1706515 1.34:4000). Western blots protein levels were quantified with Image J. Unedited blots are provided in the Supplementary Information (Supplementary Fig. 10).

**RNA isolation and qPCR.** Total RNA was extracted using an RNeasy Mini Kit (Qiagen #74104). In brief, cells were lysed in

TRIzol reagent followed by chloroform extraction and ethanol precipitation. Reverse transcription and qPCR were performed using a High Capacity cDNA reverse transcription kit (Applied Biosystems #4368814) with 2 µg per 40 µL reaction. All primer sequences are listed in Supplementary Table 3. Data was collected and analyzed using SYBR Green SuperMix Low ROX (Quanta bio # 95056-02k) with a ViiA 7 Real-Time PCR instrument (Applied Biosystems).

**Bioinformatic data.** For survival analysis, patients with metastatic CRPC who were part of the West Coast PCa Dream Team study[33] were grouped by *HSF1* mRNA so there were two groups ($n = 48$). Overall survival from the time of mCRPC diagnosis was analyzed with R and GraphPad Prism with Hazard Ratio and Log-rank (Mantel-Cox) test. RNA sequencing, Gleason score, and Grade Group were downloaded from TCGA 2015 for prostate adenocarcinoma patients[27] ($n = 236$). Patients were grouped by mRNA expression with the top quartile being considered high for either *HSF1* or *CBS* expression, so there was a *HSF1* high ($n = 61$) and *HSF1* low ($n = 175$), *CBS* high ($n = 71$) and *CBS* low ($n = 165$), and *HSF1* and *CBS* high ($n = 24$), and a *HSF1* low and *CBS* low ($n = 128$) group. Gleason score and Grade Group were analyzed by *HSF1* and *CBS* status with a Mann-Whitney U test. *HSF1* and *CBS* mRNA levels were measured in the TCGA database with 498 primary samples and 52 benign samples[27]. The GSE35988 microarray data was analyzed for *HSF1* and *CBS* mRNA levels[28]. The GPL6480 series matrix was used for *HSF1* with 49 primary tumor samples and 27 mCRPC samples. The GLP6848 series matrix was used for *CBS* with 16 benign, 10 primary tumor, and 8 mCRPC samples. *CBS* mRNA levels were measured in phs001648.v1.p1 with 44 mCRPC and 10 SCNC samples[29]. *HSF1* and *CBS* mRNA levels were analyzed with linear regression for a relationship in Beltran 2016 for the 34 patients with mCRPC[71]. There was a positive relationship seen for *HSF1* and *CBS* mRNA levels.

**Identification of HSF1 binding sites and ChIP-qPCR.** The human and mouse *CBS* gene was downloaded with the Ensembl genome browser with 10,000 base pairs before the transcription start site[35]. The gene was analyzed for HSF1 consensus binding sites[36]. Perfect HSF1 consensus binding sites or slight mismatches were found in the first exon and 3′UTR of both human and mouse *CBS* genes. Chromatin immunoprecipitation (ChIP) assays were performed as described previously[72]. C4-2 cell chromatin was crosslinked for 10 min at room temperature with 1% formaldehyde added to cell culture medium. Cell lysates were sonicated (Branson 250 #102 C) for 3 cycles of 10 seconds. Protein G-conjugated agarose beads (Santa Cruz sc-2003) were used to immunoprecipitate chromatin complexes. Four µg of ChIP grade antibodies raised against HSF1 (Abcam EP1710Y #ab52757) or the HA tag (Abcam #ab9110) on the deactivated Cas9 were used to precipitate the chromatin. After de-crosslinking and isolation of DNA with a QIAquick PCR Purification Kit, samples were analyzed with quantitative PCR was employed on genomic DNA targets relative to input with SYBR Green SuperMix Low ROX (Quanta bio # 95056-02k) and a ViiA 7 Real-Time PCR instrument (Applied Biosystems). Primers used for ChIP-qPCR are listed in Supplementary Table 3. Control IgG ChIP-qPCR assays showed low background binding to binding sites.

**Animal studies.** All animal studies were approved by the Institutional Animal Care and Use Committee (IACUC) of Duke University (A055-22-03) and Department of Defense Animal Care and use Review Office. We have complied with all relevant ethical regulations for animal use. NOD-scid IL2Rgamma[null] (NSG) mice were randomized to either be injected in the flank twice with 1 million of NCI-H660 NTC or *HSF1* and *CBS* knockout 1 in 50% stem media and 50% Matrigel (Corning #354234) with 3 NSG male mice per group, so there were six tumors per group. When the tumors reached about 100 mm³, mice were treated with 200 mg per kg doxycycline chow (Bio-Serv S3888) for the remainder of the experiment. Mice were sacrificed at 14 days because the control tumors became too large. Body weight and xenograft size was measured every Monday, Wednesday and Friday with a caliper by the same scientist throughout the experiment and calculated with size=length*(width)²*0.52 formula[48]. Tumors were fixed in 10% formalin, embedded in paraffin, and processed for IHC with HSF1 (AbCam #ab52757 at 1:200) with EnVision+ Single Reagent (Agilent Dako K400311-2) and CBS (Sigma 3E1 #WH0000875M1 at 200 ng per mL) with EnVision®+ Dual Link System-HRP (DAB+) (Agilent Dako K4065). For the SISU-102 and CH004 treatment, 24 NSG male mice were randomized into 4 groups of 6 mice. Mice were intraperitonally injected with 5 mg per kg SISU-102 in 30% Captisol, 45% saline, 20% PEG, and 5% DMSO every day until the end of the experiment. CH004 was administered at 10 mg per kg in PBS through tail vein injection by a trained technician. The mice received just two doses 48 h apart. The CH004 treatment stopped because half of the CH004 injected mice died within hours of the second injection leaving three mice per group for the CH004 only and SISU-102 and CH004 mouse groups. Mouse weight and xenograft size were measured every Monday, Wednesday, and Friday. CH004 was administered once at 7.5, 5, or 2.5 mg per kg in PBS through tail vein injection by a trained technician into 4 homozygous nu/nu nude male mice per group. Significant weight loss was observed in the 7.5 and 5 mg per kg CH004 groups, so those mice were sacrificed. However, the nude mice tolerated 2.5 mg per kg CH004. The single CH004 dose was compared to 4 nude mice treated with 7 doses of 5 mg per kg SISU-102 30% Captisol, 45% saline, 20% PEG, and 5% DMSO every day until the end of the experiment. The TRAMP FVB F1 mouse tissues were purchased from Roswell Park Cancer Institute. The Pten and Pten; Rb knockout mice were provided by the Ming Chen laboratory.

**Statistics and reproducibility.** All quantitative data were expressed as mean ± standard error. Gleason score and Grade group were analyzed as ordinal data with a Mann-Whitney U test because the Gleason score and Grade Group are not normally distributed in the PCa patient population. Statistical analysis was performed in GraphPad Prism 9 software. For experiments with two groups, two-tailed Student's *t* test was used without or without Welch's correction depending on the variance of the data. For samples with three or more groups, a one-way ANOVA was run with or without a Welch correction depending on the variance of the data. All tests were two tailed, and $P < 0.05$ was considered to be statistically significant.

The sample size for the SISU-102 treatment xenograft with NCI-H660 xenograft study was estimated by treatment of C4-2 with the HSF1 inhibitor SISU-102[14]. We calculated that our study was powered to test the primary hypothesis that there is a difference in tumor size between HSF1 inhibitor and vehicle treatment from analysis of 21 days of HSF1 inhibitor treatment in a C4-2 xenograft model. With 10 mice per group and up to 10% attrition, the two-sample *t* test with a type I error of 0.01 will have 90% power to detect a minimum effect size of about 1.95 between the two groups (Gpower 3.1). The groups for the NCI-H660 SISU-102 xenograft study were Vehicle $n = 9$ and SISU-102 $n = 10$. However, we were using both HSF1 and CBS knockout in

NCI-H660, and we had shown that there was an additive effect of HSF1 and CBS inhibition and knockout. Based on these data, we chose 3 mice per group with 2 tumors each to make 6 tumors total. This n number was chosen with the goal of keeping the number of mice used low. Mice were randomized to ensure that the average weight for each group was not different. Only male mice were used in the studies because only men develop prostate cancer. JSH was not blinded to the mice in the experiments, but the scientists measuring the xenograft size were blinded to the treatment groups.

All in vitro experiments were performed with three or more biological replicates. IncuCyte growth curves had 5 to 6 biological replicates and were repeated at least twice and had similar results. All *n* number represent biological replicates unless explicitly stated in a figure legend. All qPCR and ChIP-qPCR experiments had 3 to 4 technical replicates. Representative western blots shown in figures were repeated three times independently with similar results. High level of variance of qPCR technical replicates, wells of growth curves replicates, and metabolite profiling replicates resulted in exclusion from analysis. qPCR results with high Ct values for endogenous controls were excluded.

**Reporting summary**. Further information on research design is available in the Nature Portfolio Reporting Summary linked to this article.

## Data availability

The source data behind the graphs in the paper are available in Supplementary Data. The data generated during and/or analyzed during the current study is being stored in a LabArchives Electronic Lab Notebook, and they are available from the corresponding author Jiaoti Huang on reasonable request. Source databases to evaluate the levels of HSF1 and CBS across PCa patient types in this study include TCGA-PRAD https://portal.gdc.cancer.gov/projects/TCGA-PRAD accessed 5/14/2020, Grasso 2012 [GSE35988] and [GSE35988], Genomic Characterization of Metastatic Castration Resistant Prostate Cancer [phs001648.v1.p1] access needs to be requested from the original authors to access these data https://www.ncbi.nlm.nih.gov/projects/gap/cgi-bin/study.cgi?study_id=phs001648.v1.p1, and Molecular Basis of Neuroendocrine Prostate Cancer (Trento/Cornell/Broad 2015) [phs000909.v1.p1] access needs to be requested from the original authors to access these data https://www.ncbi.nlm.nih.gov/projects/gap/cgi-bin/study.cgi?study_id=phs000909.v1.p1. The source database to evaluate the survival of mCRPC patients by the levels of HSF1 mRNA was Genomic Characterization of Metastatic Castration Resistant Prostate Cancer [phs001648.v1.p1] access needs to be requested from the original authors to access these data https://www.ncbi.nlm.nih.gov/projects/gap/cgi-bin/study.cgi?study_id=phs001648.v1.p1. The source database to evaluate Gleason Sum and Grade Group for PCa patients was TCGA-PRAD https://portal.gdc.cancer.gov/projects/TCGA-PRAD accessed 5/14/2020.

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

## Acknowledgements

This study was supported by the Duke Clinical and Translational Science Institute Clinical and Translational Science TL1 award (TR002555), and the Department of Defense office of the Congressionally Directed Medical Research Programs Prostate Cancer Research Program Early Investigator Research Award (PC200042) awards granted to J. Spencer Hauck. We also acknowledge funding support from NIH/NCI grant (R00, CA237618) and Cancer Prevention and Research Institute of Texas (CPRIT, PR210056) Scholar in Cancer Prevention and Research award to Xia Gao. We would like to acknowledge the Martin Sjoestroem and Furong Huang of the Qianben Wang laboratory for training in survival analysis and ChIP-qPCR, respectively. The doxycycline inducible deactivated Cas9 system was kindly provided by S. Ali Shariati.

## Author contributions

J.S.H. designed all experiments, performed most experiments, analyzed the data, and wrote the manuscript. D.M. ran in vitro experiments. X.J. and M.W. measured xenograft size and treated mice. M.C. donated Pten$^{-/-}$ and Pten$^{-/-}$; Rb$^{-/-}$ tissues for IHC analysis. Y.Z. quantified tissue microarray immunohistochemistry staining intensity while being blinded to the experiment. LX cloned vectors and isolated metabolites. W.B. maintained NCI-H660 cells and troubleshot 3D aggregate growth assays. H.Q. and X.G. measured metabolite levels with LC/MS analysis. E.M., Y.H., and J.H. designed experiments and edited the manuscript.

## Competing interests

J.H. is a consultant for or owns shares in the following companies: Amgen, Artera, Kingmed Diagnostics, MoreHealth, OptraScan, York Biotechnology, Genecode, Seagen Inc. and Sisu Pharma, and received grants from Zenith Epigenetics, BioXcel Therapeutics, Inc., Dracen Pharmaceuticals and Fortis Therapeutics. All other authors declare no competing interests'.
