## [Peer Review File · Communications Biology]

Reviewers' comments:

Reviewer #1 (Remarks to the Author):

Huang et al work focused on the transsulfuration (TSS) pathway deregulated by HSF1 in prostate cancer models. The data showing an interaction and consequent regulation of CBS, a TSS critical enzyme, by HSF1 are novel and well-sustained in both in vitro and in vivo models. However, some concerns have arisen and should be addressed.

- The metabolic impact of SISU-102 on TSS and other pathways should be assessed in PC3 cell line, which shows a higher expression of HSF1 compared to C4-2 cells.
- The transsulfuration pathway is highly active to counteract oxidative stress. In your model, you do not have stress inducers. What about ROS and GSH levels in your cells? Is CBS or HSF1 targeting affects oxidative stress?
- What about CTH cystathionase expression in your cell models?
- In the tumor microenvironment, cysteine concentration is reported to be further reduced compared to that in the plasma or in the adjacent normal tissues (Kamphorst et al., 2015; Pan et al., 2016; Sullivan et al., 2019). Do you know the levels of cysteine in PCa cells and tumors along the disease progression. (i.e. CRPC)?
- The transsulfuration pathway may be hyperactivated to support tumor growth under cysteine deficiency. RPMI medium where your PCa cells are used contains cysteine. Are you able to perform HSF1/CBS expression and interaction in cysteine-free medium?
- 3D aggregate assay is used to mimic the metastatic colonization step of tumor cells. Is HSF1 and/or CBS expression high in metastatic samples or in vivo lungs tissues? The in vivo experiments ends after 20 days post-injection of tumor cells. Do you have data on their growth in late timepoints? And on their metastatic behavior? If so, what is the impact of the silencing/pharmacological approach on tumor invasiveness?
- Why did you choose NCI-H660 cell line to perform tumor xenografts? Other cell models are better investigated in vitro.

Reviewer #2 (Remarks to the Author):

This study indicates that HSF1 has relevance to prostate cancer as its expression is higher in prostate tumors vs normal prostate epithelial samples and expression is associated with poor patient outcomes. The HSF1 inhibitor SISU-102 (aka DTHIB) treated on these cell identified that loss of HSF1 activity decreased the transulfuration pathway, in particular CBS, which was also identified as a direct HSF1 target gene in PCA cells. Inhibition with small molecular or genetic knockdown/knockout of HSF1 or CBS reduced cell and tumor growth. The following comments are intended to improve the manuscript:

- 1) Authors should quantify immunoblots in Suppl Fig 3A-B and offer suggestions for why the pattern of CBS expression across these cell lines do not match the HSF1 expression from Fig 1D. Perhaps author should consider blotting for pHSF1 (S326) in Fig 1D as this represents better the active form of HSF1 rather than total HSF1 levels. Alternatively, nuclear levels of HSF1 in these cell lines would also serve as a marker for the levels of active HSF1.
- 2) The loss of HSF1 has been shown to be toxic to many cancer cells for many different purported reasons. It would be helpful to determine the importance of the decrease in transulfuration to the cell death seen with HSF1 knockdown/inhibition by determining whether exogenous CBS expression can rescue the deleterious effects of HSF1 inhibition/knockdown in one of the cell lines that were particularly sensitive to HSF1 loss.
- 3) Figure 6H-L is an elegant method of artificially blocking TF binding to ascertain the effect on endogenous gene expression.

Duke Pathology

Duke University School of Medicine

Jiaoti Huang, MD, Ph.D.
Distinguished University Professor
Johnston and West Endowed Chair
Chairman, Department of Pathology
Department of Pathology

RE: Response to reviewers comments to COMMSBIO-23-2388-T

Dear Reviewers,

We thank you for your insightful comments on our manuscript. We have performed additional experiments in response to concerns. We have included the edited manuscript with tracked changes to help you easily identify the changes.

We believe that we have satisfactorily addressed all reviewers' concerns and hope that the manuscript is now suitable for publication in Communications Biology.

Sincerely yours,

Jiaoti Huang, MD, PhD

Reviewers' comments:

Reviewer #1 (Remarks to the Author):

Huang et al work focused on the transsulfuration (TSS) pathway deregulated by HSF1 in prostate cancer models. The data showing an interaction and consequent regulation of CBS, a TSS critical enzyme, by HSF1 are novel and well-sustained in both in vitro and in vivo models. However, some concerns have arisen and should be addressed.

A) The metabolic impact of SISU-102 on TSS and other pathways should be assessed in PC3 cell line, which shows a higher expression of HSF1 compared to C4-2 cells.

Treatment of PC3 cells with the HSF1 inhibitor SISU-102 did not alter transsulfuration pathway metabolites (Supplementary Figure 4a and b). However, enrichment analysis identified that the downstream metabolites of the transsulfuration pathway taurine and glutathione were affected by SISU-102 treatment (Supplementary Figure 4a). There was a decrease in taurine and glutathione metabolites in PC3 cells after SISU-102 treatment (Supplementary Figure 4d and f). The decrease in the downstream metabolites of the transsulfuration pathway indicates that was less flux through the transsulfuration pathway due to SISU-102 treatment. C4-2 cells had decreased levels of taurine after SISU-102 treatment, but glutathione was not affected by SISU-102 (Supplementary Figure 4c and e). We included in the discussion in lines 279-287, “TSS metabolites levels were not altered from SISU-102 in PC3 cells, but taurine, an amino acid antioxidant, and glutathione, the most abundant antioxidant in mammalian cells, were decreased, indicating there was less flux through the TSS in PC3 cells treated with HSF1 inhibitor (Supplementary Figure 4a, b, d, and f)⁵⁹. C4-2 cells also had a decrease in taurine levels, but there was not a decrease in glutathione levels from SISU-102 treatment (Supplementary Figure 4c and e). We previously identified that C4-2 cells are more sensitive to SISU-102 than PC3 cells¹⁴. These data suggest that PC3 cells can compensate for SISU-102 treatment by decreasing the levels of glutathione metabolites in order to maintain steady levels of transsulfuration pathway metabolites.” All of the data from Supplementary Figure 4 is new.

Supplementary Figure 4 Transsulfuration pathway metabolites were unaffected by SISU-102 in PC3 cells, but taurine and glutathione metabolites were decreased.

(a) Homocysteine degradation was not identified from enrichment analysis with MetaboAnalyst of 5 μ M SISU-102 treated of PC3 cells, but taurine and glutathione pathway metabolites were affected (n=4). **(b)** Transsulfuration pathway metabolites were unaffected by SISU-102 treated of PC3 cells (n=4). **(c)** C4-2 cells treated with 2.5 μ M SISU-102 had decreased levels of taurine (n=4, but 1 replicate was filtered from DMSO and SISU-102 groups due to high variance). **(d)** PC3 cells treated with 5 μ M SISU-102 had decreased levels of taurine metabolites (n=4). **(e)** Glutathione levels were unaffected in C4-2 cells treated with 2.5 μ M SISU-102 (n=4, but 1 replicate was filtered from DMSO and SISU-102 groups due to high

variance). **(f)** Glutathione metabolites were decreased in PC3 cells treated with 5 μM SISU-102 (n=4). Mean \pm standard error is displayed in dot plots.

B) The transsulfuration pathway is highly active to counteract oxidative stress. In your model, you do not have stress inducers. What about ROS and GSH levels in your cells? Is CBS or HSF1 targeting affects oxidative stress?

We included in lines 247 and 248 of the discussion, “Future studies should evaluate the role of hypoxia, oxidative stress, and reactive oxygen species in the regulation of the CBS gene by HSF1.” PC3 cells decrease GSH levels to maintain steady levels of transsulfuration pathway metabolites to compensate for SISU-102 treatment (Supplemental Figure 4f). There were no conserved changes between the knockdowns and the inhibition treatments GSH in C4-2 cells.

C) What about CTH cystathionase expression in your cell models?

CTH mRNA levels were evaluated along with CBS in C4-2 HSF1 knockdown (Supplemental Figure 2c and d). CBS mRNA levels were decreased at both 3 and 7 days of doxycycline treatment without an affect on CBS levels in the non-target control. However, CTH levels were only decreased at 7 days, and CTH levels were also decreased from doxycycline treatment at 7 days in the non-target control. Therefore, CBS mRNA was directly decreased by HSF1 knockdown, while CTH appeared to be affected by doxycycline treatment. This figure was not modified from the original submission.

Supplementary Figure 2 Methionine cycle is unaffected by HSF1 inhibition and CBS is reproducibly decreased by shHSF1 but not CTH. (a) Methionine cycle metabolites are shown with

red indicating a significant increase and blue indicating a significant decrease in metabolite level. **(b)** SISU-102 treatment of C4-2 cells did not modify the levels of methionine, s-adenosyl-methionine, or s-adenyl-homocysteine (n=3). **(c, d)** Treatment of C4-2 cells with a different inducible shRNA targeting *HSF1* that was shown in the primary figures with 50 ng per mL doxycycline for 3 days **(c)** decreased CBS levels, but did not affect CTH levels (n=4). **(d)** Doxycycline treatment for 7 days decreased CBS only in the shHSF1 cells, but doxycycline treatment decreased CTH levels in both the NTC and shHSF1 cells (n=4). Mean \pm standard error is displayed in dot plots and bar graphs. CTH: cystathionine γ -lyase

D) In the tumor microenvironment, cysteine concentration is reported to be further reduced compared to that in the plasma or in the adjacent normal tissues (Kamphorst et al., 2015; Pan et al., 2016; Sullivan et al., 2019). Do you know the levels of cysteine in PCa cells and tumors along the disease progression. (i.e. CRPC)?

Cysteine levels have been reported to be decreased in the tumor microenvironment relative to plasma. Cysteine levels were measured in 262 prostate cancer samples. The levels of cysteine were increased in primary and metastatic disease compared to benign (above image from Sreekumar et al., 2009 supplemental data). We have included in lines 331-335 of the discussion, “Interestingly, unbiased metabolite profiling of 262 clinical PCa samples showed elevated levels of cysteine, homocysteine, and cystathionine in metastatic PCa and primary PCa compared to benign tissues⁶³, and increased serum levels of cysteine, homocysteine, and cystathionine independently predicted risk of early biochemical PCa recurrence and aggressiveness^{63,64}.” These data indicate that cysteine, homocysteine, and cystathionine levels are increased with the course of prostate cancer. However, we were not able to identify the ideal level of cysteine to use in in vitro studies to mimic the physiological levels of cysteine in the tumor microenvironment through literature search.

E) The transsulfuration pathway may be hyperactivated to support tumor growth under cysteine deficiency. RPMI medium where your PCa cells are used contains cysteine. Are you able to perform HSF1/CBS expression and interaction in cysteine-free medium?

To address this comment, we performed acute cysteine deprivation experiments. Acute cysteine deprivation induced CBS mRNA and protein levels in C4-2 and PC3 cells (Supplementary Fig 7a-f). Interestingly, ChIP-qPCR analysis of 0 mg/L cysteine for 48 hours compared to normal levels of cysteine (64 mg/L) showed decreased binding of HSF1 to both HSF1 binding sites in the CBS gene in C4-2 cells (Supplementary Fig 7g). These data suggest that acute cysteine

deprivation modifies the level of CBS RNA and protein independent of the action of HSF1. All of the data from Supplementary Figure 7 is new.

Supplementary Figure 7 Acute cysteine deprivation increases CBS mRNA and protein levels independent of HSF1 binding to the CBS gene. C4-2 and PC3 cells were deprived of cysteine for 48 hours before *HSF1* and *CBS* mRNA analysis (n=4) (**a, b and d, e**) and CBS protein analysis (**c and f**). The level of HSF1 binding to the 2 HSF1 binding sites on the *CBS* gene were and measured with ChIP-qPCR after 48 hours of cysteine deprivation (n=4) (**g**). Mean \pm standard error is displayed in the dot plots. The mean is only displayed in bar graphs for ChIP-qPCR experiments.

F) 3D aggregate assay is used to mimic the metastatic colonization step of tumor cells. Is HSF1 and/or CBS expression high in metastatic samples or in vivo lungs tissues? The in vivo experiments ends after 20 days post-injection of tumor cells. Do you have data on their growth in late timepoints? And on their metastatic behavior? If so, what is the impact of the silencing/pharmacological approach on tumor invasiveness?

The experiment was stopped at 20 days because the mice reached end stage criteria in our animal protocol due to the large size of the tumors. We had a pathologist evaluate major organs including liver, heart, lungs, brain, spleen, and kidney, and macroscopic metastasis was not identified by the pathologist. Organ weight was the same for all except an increase in liver weight normalized to body weight for the CBS inhibitor CH004 treated group, which was likely linked to toxicity of the CH004 inhibitor. To our knowledge NCI-H660 has not been documented to metastasize to major organs in mice after subcutaneous xenograft injection. We have added to lines 345-347 of the discussion, “Since the vast majority of SCNC biopsies are taken from metastatic sites, future studies should evaluate the efficacy of HSF1 and CBS inhibition in metastatic neuroendocrine PCa⁶⁵.”

G) Why did you choose NCI-H660 cell line to perform tumor xenografts? Other cell models are better investigated in vitro.

NCI-H660 represents small cell neuroendocrine carcinoma, the most aggressive form of prostate cancer. Patients with small cell neuroendocrine prostate carcinoma rarely live past a year after diagnosis. NCI-H660 was chosen to evaluate the efficacy of dual HSF1 and CBS ablation in a patient population that has no effective therapeutic options. We included in lines 342-345 of the discussion “NCI-H660 was selected to evaluate the effect of targeting HSF1 and CBS because we have previously shown that less aggressive in vivo models of prostate cancer including C4-2, 22Rv1, and TRAMP-C2 have a strong response to HSF1 inhibition alone¹⁴.”

Reviewer #2 (Remarks to the Author):

This study indicates that HSF1 has relevance to prostate cancer as its expression is higher in prostate tumors vs normal prostate epithelial samples and expression is associated with poor patient outcomes. The HSF1 inhibitor SISU-102 (aka DTHIB) treated on these cell identified that loss of HSF1 activity decreased the transsulfuration pathway, in particular CBS, which was also identified as a direct HSF1 target gene in PCA cells. Inhibition with small molecular or genetic knockdown/knockout of HSF1 or CBS reduced cell and tumor growth. The following comments are intended to improve the manuscript:

1) Authors should quantify immunoblots in Suppl Fig 3A-B and offer suggestions for why the pattern of CBS expression across these cell lines do not match the HSF1 expression from Fig 1D. Perhaps author should consider blotting for pHSF1 (S326) in Fig 1D as this represents better the active form of HSF1 rather than total HSF1 levels. Alternatively, nuclear levels of HSF1 in these cell lines would also serve as a marker for the levels of active HSF1.

The supplemental western blots have been quantified. We performed western blot analysis on prostate cancer lines for the level of pHSF1 (S326) with Recombinant Anti-HSF1 (phospho S326) antibody [EP1713Y] (ab76076) to determine if the levels of activated HSF1 were able to explain the discrepancy of the levels of HSF1 and CBS in the cell lines (data above). The level of pHSF1 (S326) correlated to the level of total HSF1, and it did not explain the discrepancy because LNCaP, C4-2, and 22Rv1 did not show the highest levels of active HSF1. However, we showed in the manuscript that knockdown of HSF1 decreased the level of CBS RNA and protein in the C4-2 line (Figure 6c and d), and artificially blocking HSF1 from binding to the CBS gene decreased CBS mRNA in C4-2 and 22Rv1 (Figure 6i and k). We included in lines 324-331 of the discussion, “CBS and HSF1 protein levels did not correlate in our protein analysis of PCa cell lines, possibly because there are many mechanisms by which protein levels can be regulated for these proteins including stress response^{8,11,58}. Cysteine deprivation increased CBS mRNA and protein levels in C4-2 and PC3 cells, while there was a decrease in HSF1 binding to the CBS gene (Supplementary Figure 7). These data suggest that CBS can be upregulated by cysteine

Duke Pathology

Duke University School of Medicine

deprivation independent of HSF1 activity. Future studies should investigate the regulation of CBS mRNA by HSF1 in PCa with cysteine levels and culture conditions that mimic the PCa tumor microenvironment.”

2) The loss of HSF1 has been shown to be toxic to many cancer cells for many different purported reasons. It would be helpful to determine the importance of the decrease in transsulfuration to the cell death seen with HSF1 knockdown/inhibition by determining whether exogenous CBS expression can rescue the deleterious effects of HSF1 inhibition/knockdown in one of the cell lines that were particularly sensitive to HSF1 loss.

While many groups have reported that loss of HSF1 is toxic to cancer cells, we only saw a modest increase in cell death for HSF1 inhibition or knockdown, and CBS inhibition or knockdown had a much greater impact on cell death (Figure 5). We believe that it is important to evaluate the effects of exogenous CBS in a cell line that has low basal levels of CBS protein. Many of the cell lines there were particularly sensitive to HSF1 inhibition / knockdown had higher levels of CBS protein. Therefore, we expressed exogenous CBS in PC3 cells. CBS overexpression rescued cell growth for PC3 cells treated with 2.5 and 5 μ M HSF1 inhibitor SISU-102 (Supplementary Figure 5d and e). Only the data from Supplemental Figure 5d and e is new.

Duke Pathology

Duke University School of Medicine

3) Figure 6H-L is an elegant method of artificially blocking TF binding to ascertain the effect on endogenous gene expression.

We appreciate the kind compliment from the reviewer about the elegant work to demonstrate that artificially blocking HSF1 from binding the CBS gene decreases CBS mRNA levels and increases cell death in C4-2 and 22Rv1 cells. No changes have been made to address this comment.

REVIEWERS' COMMENTS:

Reviewer #1 (Remarks to the Author):

Thanks to the authors to address all the concerns I have highlighted. In my opinion, the manuscript is now improved and suitable for publication.

Reviewer #2 (Remarks to the Author):

The revised experiments and responses of the authors was sufficient to accept the manuscript for publication.